# Urine-Based Detection of Biomarkers Indicative of Chronic Kidney Disease in a Patient Cohort from Ghana

**DOI:** 10.3390/jpm13010038

**Published:** 2022-12-24

**Authors:** Wasco Wruck, Vincent Boima, Lars Erichsen, Chantelle Thimm, Theresa Koranteng, Edward Kwakyi, Sampson Antwi, Dwomoa Adu, James Adjaye

**Affiliations:** 1Institute for Stem Cell Research and Regenerative Medicine, Medical Faculty, Heinrich Heine University, Moorenstr. 5, 40225 Düsseldorf, Germany; 2Department of Medicine & Therapeutics, University of Ghana Medical School, College of Health Sciences, Box 4236, University of Ghana, Accra P.O. Box LG 1181, Ghana; 3NHS-Clover Health Centre, Equitable House, 10 Woolich New Road, Woolich, London SE18 6AB, UK; 4Department of Child Health, School of Medical Sciences, Kwame Nkrumah University of Science and Technology, Komfo Anokye Teaching Hospital, Kumasi P.O. Box KS 9265, Ghana; 5EGA Institute for Women’s Health, University College London, 86-96 Chenies Mews, London WC1E 6HX, UK

**Keywords:** CKD, biomarkers, Ghana, urine, cytokines, VEGFA, inflammation

## Abstract

Chronic kidney disease (CKD) is a global health burden with a continuously increasing prevalence associated with an increasing incidence of diabetes and hypertension in aging populations. CKD is characterized by low glomerular filtration rate (GFR) and other renal impairments including proteinuria, thus implying that multiple factors may contribute to the etiology this disease. While there are indications of ethnic differences, it is hard to disentangle these from confounding social factors. Usually, CKD is detected in later stages of the disease when irreversible renal damage has already occurred, thus suggesting a need for early non-invasive diagnostic markers. In this study, we explored the urine secretome of a CKD patient cohort from Ghana with 40 gender-matched patients and 40 gender-matched healthy controls employing a kidney injury and a more general cytokine assay. We identified panels of kidney-specific cytokine markers, which were also gender-specific, and a panel of gender-independent cytokine markers. The gender-specific markers are IL10 and MME for male and CLU, RETN, AGER, EGFR and VEGFA for female. The gender-independent cytokine markers were APOA1, ANGPT2, C5, CFD, GH1, ICAM1, IGFBP2, IL8, KLK4, MMP9 and SPP1 (up-regulated) and FLT3LG, CSF1, PDGFA, RETN and VEGFA (down-regulated). APOA1—the major component of HDL particles—was up-regulated in Ghanaian CKD patients and its co-occurrence with APOL1 in a subpopulation of HDL particles may point to specific CKD-predisposing APOL1 haplotypes in patients of African descent—this, however, needs further investigation. The identified panels, though preliminary, lay down the foundation for the development of robust CKD-diagnostic assays.

## 1. Introduction

Chronic kidney disease (CKD) is defined by a glomerular filtration rate (GFR) < 60 mL/min per 1.73 m² or the presence of kidney damage predominantly manifested by proteinuria for 3 months or more [1,2].

Protein in urine as an indicator of kidney damage is often measured by the urine albumin-to-creatinine ratio (UACR). Proteinuria and decreased GFR directly reflect physical properties of the filter between blood and urine constituted by an endothelial layer, the glomerular basement membrane (GMB) and podocytes. This filter is coarse at the endothelial side and fine at the podocytes and works in the dimension of nanometers in addition to a negative polarity. Thus, bigger and negatively charged molecules, such as most proteins under physiological conditions, cannot traverse this barrier. In proteinuria, larger proteins, such as albumin, immunoglobulins G and M and α1-microglobulin, β2-microglobulin, correlating with the severity of histologic lesions [3], can traverse. These proteins can, as a consequence, impair the re-absorption of other smaller molecules by the proximal tubular epithelial cells and in final stages lead to toxic damage [3].

The increasing prevalence of CKD is largely influenced by the increase of diabetes and hypertension among the aging population [4,5]. Although substantial percentages of CKD patients will progress to more severe disease stages requiring dialysis or transplantation, most patients die of associated cardiovascular disease (CVD) than of end-stage renal disease (ESRD) [6]. Although diabetes is the most common cause of CKD, it is not clear why only 30% of patients with type 1 and 25 to 40% of patients with type 2 diabetes progress to nephropathy irrespective of glycemic control [4,7,8].

Ethnic differences have been reported for diabetic nephropathy, particularly in Pima Indians [4], and for focal segmental glomerulosclerosis (FSGS) in patients of African descent, who have higher frequencies of the FSGS-predisposing APOL1 G1 and G2 haplotypes, which on the other hand confer advantages against the sleeping sickness-causing parasite—Trypanosoma brucei brucei [9]. A body of literature reports gender differences in CKD [10,11,12], resulting in phenomena including gender-specific prevalence in dialysis and mortality rates. The causes of these differences have not been fully elucidated. However, differences in the levels of sex hormones and in the Renin-Angiotensin system may play major roles, as well as behavioral compliance with the dialysis treatment, medication and lifestyle restrictions.

Biomarkers for the early diagnosis of CKD are urgently needed and studies aiming at their identification have been performed using serum [13] as well as urine samples [14,15]. While there are several reports focusing on single or only few proteins [16,17,18], to our knowledge, there has been only one other study investigating CKD urine samples with a cytokine array, which, however, was on a distinct platform not specific for kidney injury markers [19]. This study identified higher levels of MMP9, MCP-1, RANTES, tissue inhibitor of metalloproteinase (TIMP) 1, TNF-alpha, vascular endothelial growth factor (VEGF), E-selectin, Fas, intercellular adhesion molecule 1, interleukin 2, matrix metalloproteinase (MMP) 2 and transforming growth factor beta and lower levels of urinary vascular cell adhesion molecule 1 and platelet-derived growth factor in CKD [19].

In this study, aimed at gender-neutral and -specific CKD biomarker identification, we investigated proteins secreted into the urine of a cohort of CKD patients and healthy controls from Ghana using a general (Human XL) assay and a kidney injury-specific cytokine assay.

## 2. Materials and Methods

Participants in the present study were recruited from two academic medical centers in urban regions of Ghana between 2012–2017. Persons with kidney disease were individuals aged 1–74 years with estimated glomerular filtration rate (eGFR) < 60 mL/min/1.73 m^2^ (creatinine based chronic kidney disease epidemiology [CKD-EPI] collaboration equation without race adjustment) [1] or albumin/creatinine ratio ≥ 3.0 mg/mmol (30 mg/g), and persons with a confirmed diagnosis of membranous glomerulonephritis, focal and segmental glomerulosclerosis/minimal change disease (FSGS/MCD) or childhood onset nephrotic syndrome. The CKD-EPI equation without the race adjustment was used in adults and the Schwartz formula [20] in children aged less than 16 years. We excluded persons with obstructive uropathy, kidney tumors, multiple myeloma, polycystic kidney disease and women who were pregnant. Healthy persons without CKD were defined as individuals with eGFR ≥ 60 mL/min/1.73 m^2^ and albumin/creatinine ratio < 3.0 mg/mmol (<30 mg/g). Random urine samples were collected from cases and controls and aliquots of 10 mLs were taken into cryovials. All participants of this study are from African ancestry. We were anticipating that ancestry might lead to some differences due to higher prevalences of genetic variants, such as the known APOL1 risk haplotype [9,21], which through regulatory networks might also affect the biomarker signatures.

### 2.1. Ethics Statement

Ethics approval was obtained locally at each clinical site. The approval number for the Case-Control study is: GHSERC: 07/03/2013. Written informed consent was obtained from all participants. Participants unable or unwilling to give consent or who were institutionalized were excluded.

### 2.2. Cytokine Assay Experiments

The urine samples were analyzed using the Human Kidney Biomarker Array (ARY019) as well as the Human XL Cytokine Array kit (ARY022B) from Research And Diagnostic Systems, Inc. (Minneapolis, MN, USA), according to the manufacturer’s protocols. For the experiment series 1, the urine samples (control and CDK) were pooled prior to analysis. In the experiment series 2, we analyzed another pool of urine samples (control and CDK).

In the experiment series 3, urine from one individual male control sample was compared to the pooled male and female CKD samples. For each membrane, 1 mL of urine was incubated overnight at 4 °C on the respective membranes. Detection was carried out using a streptavidin-HRP and the ECL Prime WesternfBlot-Detectionreagent (Merck, Darmstadt, Germany) detection reagents. The obtained signal was analyzed using FIJI/ImageJ software [22].

### 2.3. Image Analysis of Cytokine Assays

Scanned images of the cytokine assays were read into the FIJI/ImageJ software [22]. The semi-automatic grid finding was based on pre-processing via Gaussian blur (size 4) and local maxima finding as described in Steinfath et al. [23]. Local maxima were detected via the FIJI function “Find Maxima” and exported to a file in the csv format. The csv file containing the local maxima was imported into the R programming environment to detect the corners and interpolate the grid with the pre-defined size between the corners. The interpolation routine took into account alleyways between blocks within the grid and adjusted grid positions to local maxima, where existent. For positions without detected local maxima, the interpolated values were used. Grid positions were read into the FIJI Microarray Profile plugin by Bob Dougherty and Wayne Rasband (https://www.optinav.info/MicroArray_Profile.htm, accessed on 21 December 2022), which was employed to quantify the integrated densities of the spots. Grid positions were annotated with the cytokine identifiers provided in the manuals of the manufacturer (Proteome Profiler Array from R & D Systems, Human XL Cytokine Array Kit, Catalog Number ARY022B and Human Kidney Biomarker Array Kit, Catalog Number ARY019).

### 2.4. Data Analysis of Cytokine Assays

Integrated densities of the spots quantified by the FIJI Microarray plugin were imported into the R/Bioconductor [24] environment for further processing. Data were normalized via the Robust Spline Normalization from the R/Bioconductor package lumi [25]. Cytokines were considered expressed when their integrated density was significantly (*p* < 0.05) over the background spots, which were determined as spots with the minimum density plus 0.05 (max_density–min_density). Differential expression was assessed via the moderated t-test from the Bioconductor *limma* package [26]. Limma *p*-values were adjusted for the false discovery rate (FDR) by the method described by Benjamini and Hochberg [27]. Cytokines were considered as differentially expressed based on the criteria: detection *p*-value < 0.05 in at least one condition, ratio < 0.667 (down-regulated), ratio > 1.5 (up-regulated), limma-*p*-value < 0.05, FDR < 0.25. Heatmaps were generated with the function *heatmap.2* from the *gplots* package [28] using Pearson correlation as similarity measure. Bar plots were generated with the R-builtin function *barplot*. Dendrograms were produced via the R package *dendextend* [29] using Pearson correlation as the similarity measure and complete linkage as the agglomeration method.

## 3. Results

### 3.1. Patient Characteristics

Urine samples from CKD patients aged 6–71 years and control urine samples from probands with no history of CKD aged 11–68 were used (Table 1). The urine samples were collected in Ghana at the University of Ghana Medical School. Decisive criteria for the choice of CKD urine samples were the glomerular filtration rate (GFR) < 60 mL/min per 1.73 m², which is documented for CKD [29]. Furthermore, CKD is associated with an elevated albumin creatine ratio (ACR) < 300 mg/g [30]. In experiment series 1, urine samples of patients from group 1 were investigated, in experiment series 2, urine samples of patients from group 2 were investigated and in experiment series 3, urine samples of patients from group 1 were investigated.

### 3.2. Strategy for the Identification of CKD Markers

Figure 1 shows the workflow used for identifying the CKD biomarkers. In the first phase, experiment series 1 and 2 were performed on the kidney-injury cytokine assay platform. Each experiment series consisted of 10 pooled male CKD urine samples, 10 pooled female CKD urine samples, 10 pooled male healthy control urine samples and 10 pooled female healthy control urine samples. Series 1 and 2 differed by the selection of distinct individuals. In the second phase, experiment series 3 was performed on the Human cytokine XL assay platform. The experiment series consisted of 10 pooled male CKD urine samples, 10 pooled female CKD urine samples and 1 male healthy control sample.

### 3.3. Identification of CKD Biomarkers Using the Human Kidney Biomarker Array

Figure 2 shows results of experiment series 1 on the kidney cytokine assay and the identified CKD biomarkers. At the global level of the kidney cytokine expression, the data segregated (Figure 2A) into two distinct clusters of CKD (male and female) and healthy control (male and female). The heatmap (Figure 2B) and barplot (Figure 2C) show biomarkers which were found up-regulated (up) in males (M): CLU, CXCL1, IL1RN, IL10, RBP4 and SKP1, as well as biomarkers which were down-regulated (down): DPP4, EGFR, MME, MMP9 and VEGF. The heatmap (Figure 2D) and barplot (Figure 2E) show biomarkers which were found up-regulated (up) in females (F): ANPEP, ANXA5, B2M, CCL2, CCN1, CLU, CXCL16, CST3, DPP4, EGF, IL6, IL10, HAVCR1, KLK3, LCN2, MMP9, PLAU, RETN, SERPINA3, SKP1, TNFA, TNFSRF1A and TTF3, as well as biomarkers which were down-regulated (down): AGER, EGFR, FABP1 and VEGF.

Figure 3 shows the results of experiment series 2 on the Human Kidney Biomarker Array and the identified CKD biomarkers. On the global level of kidney cytokine expression, the experiments clustered heterogeneously with respect to CKD and gender (Figure 3A). The heatmap (Figure 3B) and barplot (Figure 3C) show markers which were found up-regulated (up) in males (M): ADIPOQ, AG, ANXA5, CCN1, FABP1, IL10, LCN2, MMP9, RETN, SERPINA3, TNFA, TNFSF12 and VEGF, and which were down-regulated (down): AGER, AHSG, ANPEP, CLU, CXCL16, MME and RBP4. The heatmap (Figure 3D) and barplot (Figure 3E) show markers which were found up-regulated (up) in females (F): AG, AHSG, CLU, IL1RN, MME, RETN, TNFSF12 and VCAM1, and which were down-regulated (down): ADIPOQ, AGER, ANPEP, ANXA5, CCL2, CCN1, EGFR, IL6, MMP9, PLAU and VEGF.

### 3.4. Summary of the Identified CKD Biomarkers in Experiment Series 1 and 2

Table 2 shows a summary of the CKD biomarkers identified in experiment series 1 and 2 for male CKD patients. Biomarkers identified in common in both series are underlined and in bold font. In both series, IL10 was shown to be up-regulated and MME was shown to be down-regulated in CKD.

Table 3 shows a summary of the CKD biomarkers identified in experiment series 1 and 2 for female CKD patients. Biomarkers identified in common in both series are underlined and in bold format. In both series, CLU and RETN were shown to be up-regulated and AGER, EGFR and VEGF were shown to be down-regulated in CKD.

### 3.5. Identification of CKD Biomarkers on the Human XL Cytokine Assay

After identification of CKD biomarkers on the Human Kidney Biomarker Array we set out to identify CKD-associated cytokines which include both pro-and anti-inflammatory molecules. Figure 4 shows results from experiment series 3 on the human XL cytokine assay and the identified CKD inflammation-associated biomarkers. On the global level of cytokine expression, the data segregated (Figure 4A) into two clusters of CKD (male and female) and healthy control (male). The heatmap (Figure 4B) and barplot (Figure 4C) show biomarkers which were found to be up-regulated (up) or down-regulated (down) in males (M). The heatmap (Figure 4D) and barplot (Figure 4E) show markers which were found up-regulated (up) or down-regulated in females (F). Interestingly, we found a large overlap between up-and down-regulated cytokines between both genders. Some biomarkers that were up-regulated in experiment series 3 overlapped between females (F) and males (M) in CKD: APOA1 (up), ANGPT2 (up), C5 (up), CFD (up), GH1 (up), ICAM1 (up), IGFBP2 (up), IL8 (up), KLK4 (up), MMP9 (up) and SPP1 (up). Other markers that were down-regulated in experiment series 3 overlapped between female (F) and male (M) in CKD: FLT3LG (down), CSF1 (down), PDGFA (down), RETN (down) and VEGFA (down).

Table 4 shows a summary of CKD markers identified in experiment series 3 for male and female CKD patients and independent of gender, taking the mean of male and female values. Markers identified as overlapping in males and females are underlined and in bold format. This summary table highlights the large overlap between male and female CKD markers on the human XL cytokine platform.

### 3.6. Protein Interaction Network

Based on the identified differentially regulated cytokines, we set out to identify and characterize a protein interaction network involved in the inflammatory pathophysiology of CKD. Figure 5A shows the Protein interaction network generated via the STRING online tool [31]. Via the STRING tool, enriched disease associations with *Leukostasis* and *Artery Disease* (Figure 5C) were determined and a table of enriched Gene ontology Biological Processes was created (Figure 5D, top 25 terms sorted by strength is shown). These pointed to the inflammatory response and included the highlighted term *Regulation of kidney development* (strength = 1.8, false discovery rate = 0.0024). The proteins associated with the regulation of kidney development (PDGFA, MMP9, VEGFA) are highlighted in the network in Figure 5B.

## 4. Discussion

In this two-phase analysis of cytokines in the urine of CKD patients and healthy individuals from Ghana, we identified urine-based cytokine biomarkers for CKD. In the first phase, which was performed on a Human Kidney Biomarker Array, we found variability between males and females and also between the two experimental series, which consisted of pooled urine samples from distinct individuals. This variability is a reflection of heterogeneity within biomarkers associated with CKD, which comprises a broad spectrum of distinct diseases, such as diabetic nephropathy and focal segmental glomerulosclerosis. Furthermore, there is much diversity in the genetic background of CKD and recent findings on biomarkers associated with hitherto neglected areas, such as telomeres, CNVs, mtDNA variants and sex chromosomes [32], which may gain attention in the future. This may also help to explain the high heredity estimated at 30–75%, which at the moment is incompletely understood [32]. A further aspect of heterogeneity is found in the definitions of CKD, which on the one hand have to cope with the complexity of the disease, but on the other hand, lead to strongly deviating assessments of CKD progression [33]. Among the kidney-associated cytokines, we identified as variable between genders and experimental series LCN2 (alias NGAL; up-regulated in male in experiment series 1 and in female in experiment series 2, not significant in the others), which should be highlighted. NGAL had been established as marker of acute kidney injury, but several problems, including its unpredictable release, have led to increasing concerns about its diagnostic value [34].

For the sake of robustness, only biomarkers regulated in the same direction in both experiment series are listed. The CKD biomarkers in male are: IL10 (Interleukin-10, up-regulated) and MME (Membrane metalloendopeptidase, down-regulated). Sinuani et al. reported that IL10, through increased proliferation of mesangial cells and mediated by several other cytokines, induces progression of renal failure [35]. Dedicated single nucleotide polymorphisms (SNPs) in the MME gene have been associated with a higher risk for diabetic nephropathy in female diabetes type 1 patients [36]. The CKD biomarkers in females are: CLU (Clusterin), RETN (Resistin, up-regulated), AGER (advanced glycosylation end-product specific receptor, alias RAGE), EGFR (epidermal growth factor receptor) and VEGFA (vascular endothelial growth factor A, alias VEGF, down-regulated). CLU has been reported to be elevated in kidney disease [37,38], although Guo et al. found that CLU deficiency exacerbates renal inflammation and tissue fibrosis after ischemia-reperfusion injury in mice [39]. For Resistin, there are reports on elevated levels in CKD, which are associated with decreased glomerular filtration rate and inflammation [40]. AGER/RAGE was reported to be elevated in the serum of CKD patients [41]. Up-regulation of EGFR has been described for CKD, but EGFR inhibition in models of acute kidney injury (AKI) may also have deleterious effects [42].

In the second phase, we used a more general cytokine (human XL) assay and found a high percentage of cytokine biomarkers overlapping between both genders. The CKD biomarkers regulated in the same direction between male and female are: APOA1, ANGPT2, C5, CFD, GH1, ICAM1, IGFBP2, IL8, KLK4, MMP9 and SPP1 (up-regulated) and FLT3LG, CSF1, PDGFA, RETN and VEGFA (down-regulated).

Interestingly, APOA1 is connected to APOL1, which has been associated with FSGS in haplotypes carried by patients of African descent. APOL1 is bound to HDL particles, of which APOA1 is the major protein component, but only 10% of APOA1-containing HDL particles have APOL1 [21]. We still need to confirm if these APOL1-positive HDL particles play a special role in CKD. In comparison to the first phase, where we saw strong gender-specific differences, in the second phase, we identified nearly the same markers regulated in the same direction for both males and females. The large number of gender-specific differences found on the Human Kidney Biomarker Array in the first phase is in line with reports of gender differences in kidney function [43] and kidney disease [11]. It is tempting to speculate that the more general cytokines measured on the human XL assay reflect more gender-independent inflammatory processes which precede CKD.

Cytokines detected as differentially regulated on the Human XL cytokine assay as well as the Human Kidney Biomarker Array were RETN (Resistin, down-regulated on the Human XL cytokine assay), MMP9 (metalloproteinase 9, up-regulated on the Human XL cytokine assay) and VEGFA (down-regulated on the Human XL cytokine assay). VEGFA was regulated in the same direction on the Human Kidney Biomarker Array (down), while MMP9 was up-regulated in males in series 2 and females in series 1 and down-regulated in males in series 1 and females in series 2; however, RETN, in contrast to the XL assay, was up-regulated. As mentioned above for RETN, there are reports on elevated levels in CKD [40]. MMP9 is described to be up-regulated in early stages of the disease and down-regulated in later stages [44]. In our cohorts, this might reflect the predominantly late disease stages in the distinct cohorts. For VEGFA, there are conflicting data on the regulation in diabetic nephropathy going from up-regulation in rat models and also diabetic patients [45], to no effect [46], to down-regulation when diabetic nephropathy leads to glomerusclerosis [45]. For glomerusclerosis and tubulorinterstitial fibrosis down-regulation has been reported [45]. 

We compared our detected biomarkers with association loci from GWAS and found that VEGFA, which we detected as a biomarker, had been identified in GWAS for an association with eGFR [47] and also in a further study by the CKD consortium for an association with CKD, which, while not having a significant *p*-value, nevertheless pointed in the same direction [48]. Furthermore, we found that CST3, which we found up-regulated in females on the Human Kidney biomarker array (Figure 2E), was detected to be associated with eGFR via GWAS by Köttgen et al. [47].

We anticipate that our detected biomarkers may foster the mechanistic understanding of the development of CKD and further help to enable an earlier detection of the disease and in cases not detected by the conventional markers. CKD diagnosis mainly relies on GFR estimation indicating kidney dysfunction or albuminuria indicating kidney damage [49]. GFR estimations are hampered by limitations of creatinine [50]. Albumin does not work as a marker in the non-albuminuric cases highly prevalent in non-diabetic CKD [17,51,52], but is also common in diabetic CKD [16,49,53].

## 5. Conclusions

We conclude that our two-phase cytokine analysis of urine samples from CKD patients and healthy controls from Ghana has revealed panels of kidney-specific cytokine biomarkers which are also gender-specific and a panel of gender-independent cytokine markers. The gender-specific markers are IL10 and MME for males and CLU, RETN, AGER, EGFR and VEGFA for females. The gender-independent cytokine markers were APOA1, ANGPT2, C5, CFD, GH1, ICAM1, IGFBP2, IL8, KLK4, MMP9 and SPP1 (up-regulated) and FLT3LG, CSF1, PDGFA, RETN and VEGFA (down-regulated).

## Figures and Tables

**Figure 1 jpm-13-00038-f001:**
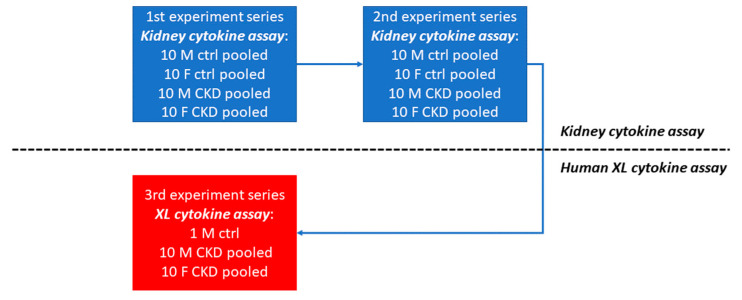
Workflow for the identification of CKD biomarkers. The pipeline employed the Human Kidney Biomarker Array in phase 1 of experiment series 1 and 2 and the human XL cytokine assay in phase 2 with experiment series 3. Pools of 10 male and 10 female urine samples were used for CKD patients and healthy controls.

**Figure 2 jpm-13-00038-f002:**
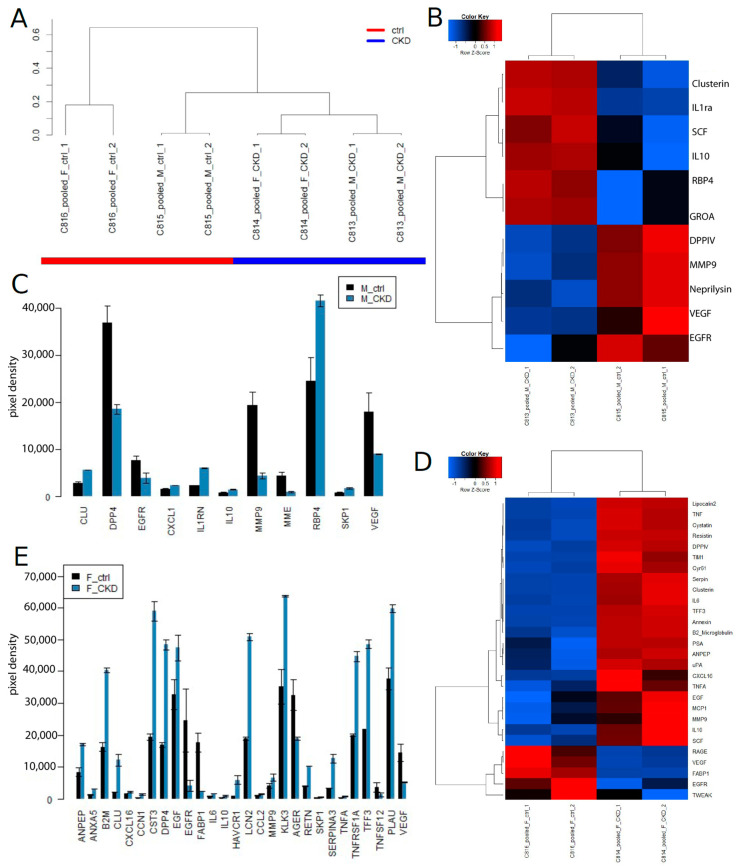
CKD biomarkers identified on the Human Kidney Biomarker Array in experiment se-ries 1. (**A**) Experiments cluster into CKD and healthy control based on the global kidney cytokine expression. (**B**) Heatmap and (**C**) barplot of markers in males (M) that were up-regulated (up): CLU, CXCL1, IL1RN, IL10, RBP4 and SKP1, and down-regulated (down): DPP4, EGFR, MME, MMP9 and VEGF. (**D**) Heatmap and (**E**) barplot of markers in females (F) that were up-regulated (up): ANPEP, ANXA5, B2M, CCL2, CCN1, CLU, CXCL16, CST3, DPP4, EGF, IL6, IL10, HAVCR1, KLK3, LCN2, MMP9, PLAU, RETN, SERPINA3, SKP1, TNFA, TNFSRF1A and TTF3, and down-regulated (down): AGER, EGFR, FABP1 and VEGF.

**Figure 3 jpm-13-00038-f003:**
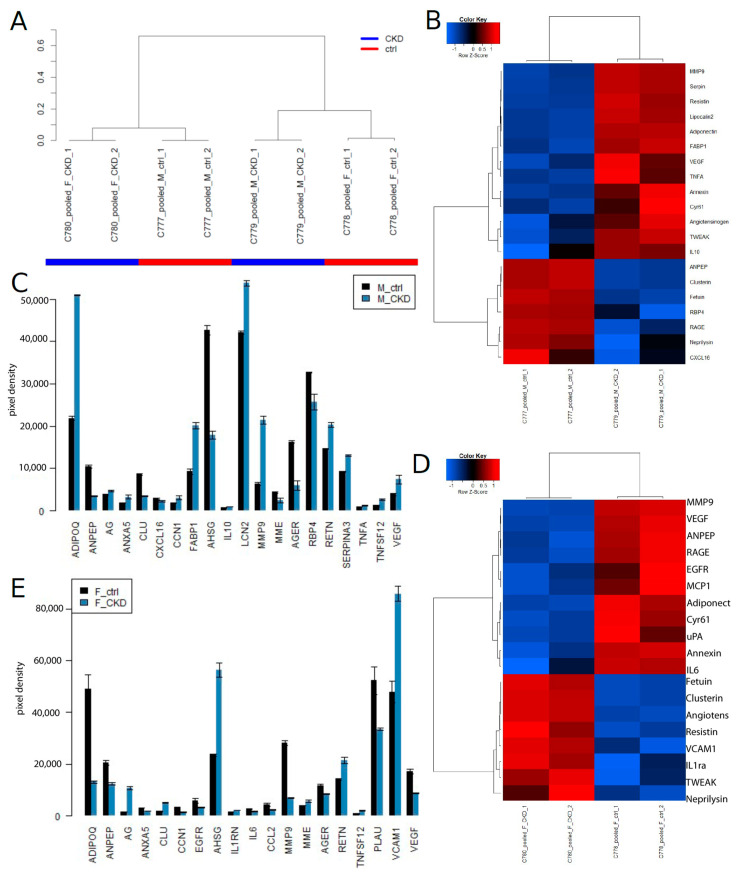
CKD biomarkers identified on the Human Kidney Biomarker Array in experiment series 2. (**A**) experiments cluster heterogeneously based on the global kidney cytokine expression. (**B**) Heatmap and (**C**) barplot of biomarkers in males (M) that were up-regulated (up): ADIPOQ, AG, ANXA5, CCN1, FABP1, IL10, LCN2, MMP9, RETN, SERPINA3, TNFA, TNFSF12 and VEGF, and down-regulated (down): AGER, AHSG, ANPEP, CLU, CXCL16, MME and RBP4. (**D**) Heatmap and (**E**) barplot of biomarkers in females (F) that were up-regulated (up): AG, AHSG, CLU, IL1RN, MME, RETN, TNFSF12 and VCAM1, and down-regulated (down): ADIPOQ, AGER, ANPEP, ANXA5, CCL2, CCN1, EGFR, IL6, MMP9, PLAU and VEGF.

**Figure 4 jpm-13-00038-f004:**
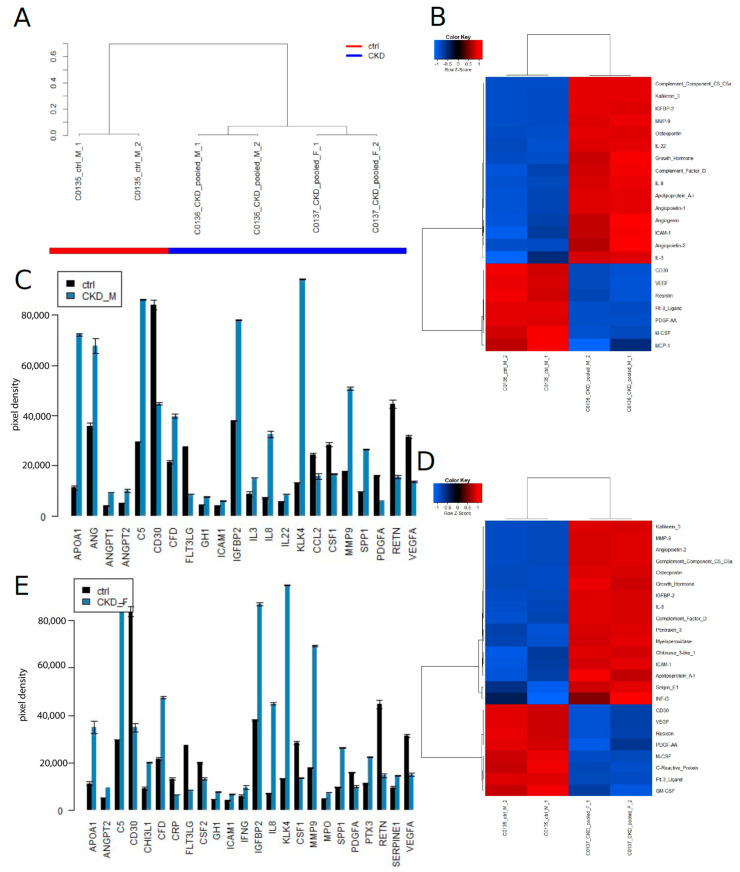
CKD biomarkers identified on the Human XL cytokine assay in experiment series 3 show significant overlap between males and females. (**A**) Data cluster into CKD and healthy control based on global cytokine expression. (**B**) Heatmap and (**C**) barplot of markers in males (M) that were up-regulated (up) or down-regulated (down). (**D**) Heatmap and (**E**) barplot of markers in females (F) that were up-regulated (up) or down-regulated (down). Markers in experiment series 3 that overlapped between females (F) and males (M) that were up-regulated in CKD: APOA1 (up), ANGPT2 (up), C5 (up), CFD (up), GH1 (up), ICAM1 (up), IGFBP2 (up), IL8 (up), KLK4 (up), MMP9 (up) and SPP1 (up). Markers in experiment series 3 that overlapped between females (F) and males (M) that were down-regulated in CKD: FLT3LG (down), CSF1 (down), PDGFA (down), RETN (down) and VEGFA (down).

**Figure 5 jpm-13-00038-f005:**
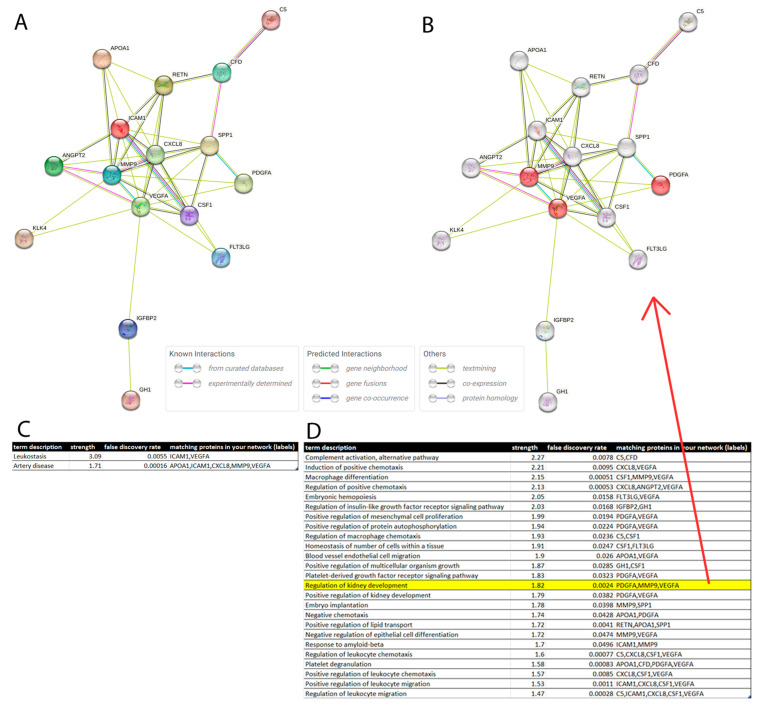
CKD biomarkers identified on the Human XL cytokine assay interact in a network regulating the inflammatory response affecting kidney development. (**A**) STRING protein interaction network (PPI) of CKD biomarkers identified on the Human XL cytokine assay. (**B**) PPI highlighting proteins involved in the regulation of kidney development. (**C**) Disease associations found enriched by the STRING tool are *Leukostasis* and *Artery Disease*. (**D**) Gene ontology Biological Processes found enriched point to the inflammatory response and include *the regulation of kidney development*.

**Table 1 jpm-13-00038-t001:** Characteristics of the patient cohort and the healthy individual cohort utilized in the cytokine assay analyses.

**Group 1**															
**Control female**				**Control male**				**CKD female**				**CKD male**			
**Age**	**eGFR (mL/min per 1-73 m²)**	**number**	**Albumine creatine ratio (g/mmol)**	**Age**	**eGFR (mL/min per 1-73 m²)**	**number**	**Albumine creatine ratio (g/mmol)**	**Age**	**eGFR (mL/min per 1-73 m²)**	**number**	**Albumine creatine ratio (g/mmol)**	**Age**	**eGFR (mL/min per 1-73 m²)**	**number**	**Albumine creatine ratio (g/mmol)**
39	169.45	52815	0.68	11	70.77	52803	0.4	69	3	52866	107.25	24	84.05	52852	123.34
39	94.85	21206	1.11	26	153.98	52833	1.6	28	29	21245	8.27	41	123.38	52840	10.61
46	131.85	52855	0.85	46	92.64	52856	0.95	65	49	52838	1.67	28	153.97	52829	1925.38
47	145.27	52805	0.47	47	120.56	52839	1.06	49	51	52973	30.68	25	181.34	52830	17.89
20	134.71	21227	0.81	47	85.62	52839	1.06	38	75	52845	13.79	48	11.04	52842	2.69
24	74.53	21246	0.51	57	160.36	52777	1.03	39	92	21240	119.93	47	30.99	52875	0.91
25	113.26	21216	0.19	57	136.93	52823	0.48	25	93	52854	3.76	48	11.04	52842	2.69
70	134.61	52778	0.3	66	145.09	52818	0.67	35	99	21244	11.23	60	22.58	52828	87.79
51	150.26	52757	0.32	67	83.58	21241	0.36	46	114	52827	141.83	60	54.94	52862	3.05
50	119.12	21225	0.35	68	102.38	52861	1.71	60	123	52871	82.99	65	4.95	52832	0.54
**Group 2**															
**Control female**				**Control male**				**CKD female**				**CKD male**			
**Age**	**eGFR (mL/min per 1-73 m²)**	**number**	**Albumine creatine ratio (g/mmol)**	**Age**	**eGFR (mL/min per 1-73 m²)**	**number**	**Albumine creatine ratio (g/mmol)**	**Age**	**eGFR (mL/min per 1-73 m²)**	**number**	**Albumine creatine ratio (g/mmol)**	**Age**	**eGFR (mL/min per 1-73 m²)**	**number**	**Albumine creatine ratio (g/mmol)**
33	185.43	52768	1.32	17	72.7	21210	0.26	65	1	52865	0.68	22	157.26	52843	2.55
4	85.97	52811	0.71	18	83.16	21223	2.02	22	3	52877	0.62	24	19.68	52844	0.97
14	92.56	21213	0.52	21	165.3	21232	1.24	71	11	52834	0.91	28	5.37	52864	5.03
16	75.12	21214	0.37	24	153.06	21212	0.26	7	14	21237	0.68	33	194.92	52831	10
16	68.11	52812	2.51	33	217.41	52804	1.21	30	17	52859	1.04	35	24.3	52873	1.2
18	78.64	21231	0.86	37	151.17	21235	0.88	50	26	21238	0.6	37	13.96	52870	1.08
20	129.93	21229	0.76	48	123.11	21243	1.14	71	35	52867	0.53	39	5.92	21248	2.08
27	140.59	21218	1.92	49	68.92	52728	0.52	62	41	21249	2.78	39	8.5	52879	0.57
62	153.39	52825	1.84	59	85.61	52759	0.89	28	46	52857	46.4	44	5.15	52835	4.33
21	91.94	21217	0.78	63	87.59	52772	1.36	35	17	52858	7.2	59	3.07	52869	0.51

**Table 2 jpm-13-00038-t002:** CKD markers identified in male CKD patients in experiment series 1 and 2.

Cytokine	CKD_M_ExpSeries1	CKD_M_ExpSeries2
ADIPOQ		up
AG		up
AGER		down
AHSG		down
ANPEP		down
ANXA5		up
CCN1		up
CLU	up	down
CXCL1	up	
CXCL16		down
DPP4	down	
EGFR	down	
FABP1		up
IL1RN	up	
** IL10 **	** up **	** up **
LCN2		up
** MME **	** down **	** down **
MMP9	down	up
RBP4	up	down
RETN		up
SERPINA3		up
SKP1	up	
TNFA		up
TNFSF12		up
VEGF	down	up

**Table 3 jpm-13-00038-t003:** CKD markers identified in female CKD patients in experiment series 1 and 2.

Cytokine	CKD_F_ExpSeries1	CKD_F_ExpSeries2
ADIPOQ		**down**
AG		**up**
** AGER **	** down **	** down **
AHSG		**up**
ANPEP	**up**	**down**
ANXA5	**up**	**down**
B2M	**up**	
CCL2	**up**	**down**
CCN1	**up**	**down**
** CLU **	** up **	** up **
CST3	**up**	
CXCL16	**up**	
DPP4	**up**	
EGF	**up**	
** EGFR **	** down **	** down **
FABP1	**down**	
IL1RN		**up**
IL6	**up**	**down**
IL10	**up**	
HAVCR1	**up**	
KLK3	**up**	
LCN2	**up**	
MME		**up**
MMP9	**up**	**down**
PLAU	**up**	**down**
** RETN **	** up **	** up **
SERPINA3	**up**	
SKP1	**up**	
TNFA	**up**	
TNFSF12		**up**
TNFRSF1A	**up**	
TTF3	**up**	
VCAM1		**up**
** VEGF **	** down **	** down **

**Table 4 jpm-13-00038-t004:** CKD-associated cytokine biomarkers identified as common and specific between males and females, based on experiment series 3.

Cytokine	CKD_mean_M_F_ExpSeries3	CKD_M_ExpSeries3	CKD_F_ExpSeries3
** APOA1 **	** up **	** up **	** up **
ANG		**up**	
ANGPT1		**up**	
** ANGPT2 **	** up **	** up **	** up **
** C5 **	** up **	** up **	** up **
CD3	**down**		**down**
CHI3L1	**up**		**up**
** CFD **	** up **	** up **	** up **
CRP	**down**		**down**
** FLT3LG **	** down **	** down **	** down **
CSF2	**down**		**down**
** GH1 **	** up **	** up **	** up **
** ICAM1 **	** up **	** up **	** up **
IFNG	**up**		**up**
** IGFBP2 **	** up **	** up **	** up **
IL3		**up**	
** IL8 **	** up **	** up **	** up **
IL22		**up**	
** KLK4 **	** up **	** up **	** up **
CCL2		**down**	
** CSF1 **	** down **	** down **	** down **
** MMP9 **	** up **	** up **	** up **
MPO	**up**		**up**
** SPP1 **	** up **	** up **	** up **
** PDGFA **	** down **	** down **	** down **
PTX3	**up**		**up**
** RETN **	** down **	** down **	** down **
SERPINE1	**up**		**up**
** VEGFA **	** down **	** down **	** down **

## Data Availability

Datasets for the analysis of cytokine assays were generated during the current study. The datasets are available in the Appendix A data of this work.

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
