# Peer review of "Urine-Based Detection of Biomarkers Indicative of Chronic Kidney Disease in a Patient Cohort from Ghana"

_jpm, 2022, doi:10.3390/jpm13010038_

Round 1

Reviewer 1 Report

Dear Authors, 

 the main question is about novel urinary biomarkers in CKD different Thank creatinine or KIM 1.  It is not original topic, but i consider relevant  the biomarkers considered.  it add the potential role of different cytokines like IL-10.  the Authors should assess over time (6-12 months) the reliability of the markers assessed.   i think the conclusions are not conclusive consider the main aim of the study, since the markers should be assessed over time.   i consider the references appropriated.  i have no additional comments on tables  and figures.

i appreciated your efforts and your manuscript. Hope to check it on following edition.

Author Response

Reviewer #1

The reviewer wrote:

the main question is about novel urinary biomarkers in CKD different Thank creatinine or KIM 1.  It is not original topic, but i consider relevant  the biomarkers considered.  it add the potential role of different cytokines like IL-10.  the Authors should assess over time (6-12 months) the reliability of the markers assessed.   i think the conclusions are not conclusive consider the main aim of the study, since the markers should be assessed over time.   i consider the references appropriated.  i have no additional comments on tables  and figures.

 i appreciated your efforts and your manuscript. Hope to check it on following edition.

Our reply:

We thank the reviewer for the constructive comments and agree that it would be very interesting to have the reliabilty of the markers assessed over time. However, this was a pilot study which should lay down the foundation for further studies which can address this very important issue. This was a cross sectional study, so we have urine samples for one time point only.

Reviewer 2 Report

Review of jpm-2057881

Urine-based detection of biomarkers indicative of chronic kidney disease in a patient cohort from Ghana

Wasco Wruck, Vincent Boima, Lars Erichsen, Chantelle Thimm, Theresa Koranteng, Edward Kwakyi, Sampson Antwi, Dwomoa Adu and James Adjaye

General comments

This manuscript addresses the urine secretome of a CKD patient cohort from Ghana. The topic addressed is interesting. But I think that the results are not well discussed. The introduction, result and discussion part must be improved with more explanations and discussions. I cannot recommend the acceptance of the present paper in this form for publication.

Specific comments

1, In abstract, the authors had better to include the sample size.

2, In introduction, the authors would include the same kind of reports which have already performed or known previously.

3, In methods, were the participants from African ancestry? Do the authors think biomarkers may differ by ancestry?

4, In methods, were the patients biopsied or clinically diagnosed? Pediatric (under 18 years old) patients should not be applied CKD-EPI to calculate eGFR. Please apply appropriate equation.

5, In methods, I did not follow the random urine samples. Were urine samples collected multiple times for each participant? I believe that urinary biomarkers may vary depending on the status and condition of treatment. The authors need to set the same conditions or average several times.

6, In methods, line 94 shows “first round, second round and third round” and line 145 shows “Series 1 and series 2 and series 3”. Do these mean the same thing?

7, Have the authors checked the APOL1 risk haplotype for the participants?

8, Were there any differences in background disease? The authors would include the patients’ characteristics in the table.

9, The authors addressed the gender specific biomarkers. The authors would include the motivation of checking the sex differences in introduction.

10, Currently, GWAS of CKD or eGFR were performed and detected lots of association loci. Did the authors find any association between these loci and biomarkers?

11, Do the authors have any thoughts on how to use these biomarkers to detect patients with CKD? What is the difference between testing for urinary albumin or proteinuria and using this biomarker?

12, In table 1, the number of age and eGFR of CKD female seems to be flipped.

Author Response

Reviewer #2

The reviewer wrote:

General comments

This manuscript addresses the urine secretome of a CKD patient cohort from Ghana. The topic addressed is interesting. But I think that the results are not well discussed. The introduction, result and discussion part must be improved with more explanations and discussions. I cannot recommend the acceptance of the present paper in this form for publication.

Our reply:

We have revised the introduction, result and discussion chapters of the manuscript according to the specific comments and are convinced that the revisions have improved the manuscript significantly.

Specific comments

The reviewer wrote:

1, In abstract, the authors had better to include the sample size.

Our reply:

We have now included the sample size in the abstract:

“In this study, we explored the urine secretome of a CKD patient cohort from Ghana with 40 gender-matched patients and 40 gender-matched healthy controls employing a kidney-injury and a more general cytokine assay.”

The reviewer wrote:

2, In introduction, the authors would include the same kind of reports which have already performed or known previously.

Our reply:

We now mention similar reports in the introduction:

“While there are several reports focussing on single or only few proteins [13], [14], [15] to our knowledge there has been only one other study investigating CKD urine samples with a cytokine array which however was on a distinct platform not specific for kidney injury markers [16]. This study identified higher levels of MMP9, MCP-1, RANTES, tissue inhibitor of metalloproteinase (TIMP) 1, TNF-alpha, vascular endothelial growth factor (VEGF), E-selectin, Fas, intercellular adhesion molecule 1, interleukin 2, matrix metalloproteinase (MMP) 2 and transforming growth factor beta and lower levels of urinary vascular cell adhesion molecule 1 and platelet-derived growth factor in CKD [16] .”

The reviewer wrote:

3, In methods, were the participants from African ancestry? Do the authors think biomarkers may differ by ancestry?

Our reply:

We now mention that all participants are from African ancestry in the methods and that we were anticipating that some biomarkers may differ by ancestry:

“All participants of this study are from African ancestry. We were anticipating that ancestry might lead to some differences due to higher prevalences of genetic variants such as the known APOL1 risk haplotype [9], [17] which through regulatory networks might also affect the biomarker signatures.”

The reviewer wrote:

4, In methods, were the patients biopsied or clinically diagnosed? Pediatric (under 18 years old) patients should not be applied CKD-EPI to calculate eGFR. Please apply appropriate equation.

Our reply:

The diagnosis of CKD was based on an GFR of<60.  The CKD-EPI equation without the race adjustment was used in adults and the Schwartz formula in children aged less than 16 years. (Schwartz GJ, Munoz A, Schneider MF, Mak RH, Kaskel F, Warady BA, et al. New equations to estimate GFR in children with CKD. Journal of the American Society of Nephrology. 2009;20(3):629-37)

We have added this to “2. Materials and Methods”:

“The CKD-EPI equation without the race adjustment was used in adults and the Schwartz formula [20] in children aged less than 16 years.”

The reviewer wrote:

5, In methods, I did not follow the random urine samples. Were urine samples collected multiple times for each participant? I believe that urinary biomarkers may vary depending on the status and condition of treatment. The authors need to set the same conditions or average several times.

Our reply:

We collected urine samples on only one occasion as this was a cross sectional study.

The reviewer wrote:

6, In methods, line 94 shows “first round, second round and third round” and line 145 shows “Series 1 and series 2 and series 3”. Do these mean the same thing?

Our reply:

We have replaced “round” by “series” in methods:

“For the experiment series 1 the urine samples (control and CDK) were pooled prior to analysis. In the experiment series 2 we analyzed another pool of urine samples (control and CDK).

In the experiment series 3 …”

The reviewer wrote:

7, Have the authors checked the APOL1 risk haplotype for the participants?

Our reply:

No.

The reviewer wrote:

8, Were there any differences in background disease? The authors would include the patients’ characteristics in the table.

Our reply:

We recruited subjects with CKD as defined by an GFR<60 ml/min/1.73 m2 .  We did not further characterize the cause of the CKD.

The reviewer wrote:

9, The authors addressed the gender specific biomarkers. The authors would include the motivation of checking the sex differences in introduction.

Our reply:

There are many studies reporting gender differences for CKD. We have now added this motivation and cite some of the studies in the introduction:

“A body of literature report gender differences in CKD [10], [11], [12] resulting in phenomena including gender-specific prevalence in dialysis and mortality rates. The causes of these differences have not been fully elucidated. However, differences in the levels of sex hormones and in the Renin-Angiotensin system may play major roles as well as behavioral compliance with the dialysis treatment, medication and lifestyle restrictions.”

The reviewer wrote:

10, Currently, GWAS of CKD or eGFR were performed and detected lots of association loci. Did the authors find any association between these loci and biomarkers?

Our reply:

We added the comparison to GWAS-detected loci to the discussion (we had missed CST3 in the legend and in the text but now corrected this):

 “We compared our detected biomarkers with association loci from GWAS and found that VEGFA which we detected as biomarker had been identified in GWAS for an association with eGFR [47] and also in a further study by the CKD consortium for an association with CKD which while not having a significant p-value nevertheless pointed in the same direction [48]. Furthermore, we found that CST3 which we found up-regulated in females on the Human Kidney biomarker array (Figure 2E) was detected to be associated with eGFR via GWAS by Köttgen et al. [47]. ”

The reviewer wrote:

11, Do the authors have any thoughts on how to use these biomarkers to detect patients with CKD? What is the difference between testing for urinary albumin or proteinuria and using this biomarker?

Our reply:

We elaborate on how these biomarkers may help to detect CKD also in cases without albuminuria in the Discussion.

“We anticipate that our detected biomarkers may foster the mechanistic understanding of the development of CKD and further help to enable an earlier detection of the disease and in cases not detected by the conventional markers. CKD diagnosis mainly relies on GFR estimation indicating kidney dysfunction or albuminuria indicating kidney damage [49]. GFR estimations are hampered by limitations of creatinine [50]. Albumin does not work as marker in the non-albuminuric cases highly prevalent in non-diabetic CKD [17], [51], [52] but also common in diabetic CKD [53], [49], [16] .“

The reviewer wrote:

12, In table 1, the number of age and eGFR of CKD female seems to be flipped.

Our reply:

We have changed the table according to the reviewer’s comment.

Round 2

Reviewer 2 Report

I appreciate the authors for their collaboration.

The manuscript has been much improved.